# Improvement of image quality and assessment of respiratory motion for hepatocellular carcinoma with portal vein tumor thrombosis using contrast-enhanced four-dimensional dual-energy computed tomography

**Shingo Ohira**[1,2]\*, **Naoyuki Kanayama**[1], **Kentaro Wada**[1], **Toshiki Ikawa**[1], **Takero Hirata**[1], **Noriko Kishi**[1,3], **Tsukasa Karino**[1], **Hayate Washio**[1], **Yoshihiro Ueda**[1], **Masayoshi Miyazaki**[1], **Masahiko Koizumi**[2], **Teruki Teshima**[1]

**1** Department of Radiation Oncology, Osaka International Cancer Institute, Osaka, Japan, **2** Department of Medical Physics and Engineering, Osaka University Graduate School of Medicine, Suita, Japan, **3** Department of Radiation Oncology and Image-Applied Therapy, Graduate School of Medicine, Kyoto University, Kyoto, Japan

\* oohira-si@mc.pref.osaka.jp

**Data Availability Statement:** All relevant data are within the manuscript.

## Abstract

To assess the objective and subjective image quality, and respiratory motion of hepatocellular carcinoma with portal vein tumor thrombosis (PVTT) using the contrast-enhanced four-dimensional dual-energy computed tomography (CE-4D-DECT). For twelve patients, the virtual monochromatic image (VMI) derived from the CE-4D-DECT with the highest contrast to noise ratio (CNR) was determined as the optimal VMI (O-VMI). To assess the objective and subjective image quality, the CNR and five-point score of the O-VMI were compared to those of the standard VMI at 77 keV (S-VMI). The respiratory motion of the PVTT and diaphragm was measured based on the exhale and inhale phase images. The VMI at 60 keV yielded the highest CNR ($4.8 \pm 1.4$) which was significantly higher ($p = 0.02$) than that in the S-VMI ($3.8 \pm 1.2$). The overall image quality ($4.0 \pm 0.6$ vs $3.1 \pm 0.5$) and tumor conspicuity ($3.8 \pm 0.8$ vs $2.8 \pm 0.6$) of the O-VMI determined by three radiation oncologists was significantly higher ($p < 0.01$) than that of the S-VMI. The diaphragm motion in the L-R ($3.3 \pm 2.5$ vs $1.2 \pm 1.1$ mm), A-P ($6.7 \pm 4.0$ vs $1.6 \pm 1.3$mm) and 3D ($8.8 \pm 3.5$ vs $13.1 \pm 4.9$ mm) directions were significantly larger ($p < 0.05$) compared to the tumor motion. The improvement of objective and subjective image quality was achieved in the O-VMI. Because the respiratory motion of the diaphragm was larger than that of the PVTT, we need to be pay attention for localizing target in radiotherapy.

**Funding:** This study was supported by JSPS KAKENHI Grant (Grant-in-Aid for Young Scientists 19K17285). The funders had no role in study design, data collection and analysis, decision to publish, or preparation of the manuscript.

**Competing interests:** The authors have declared that no competing interests exist.

## Introduction

Hepatocellular carcinoma (HCC) is the second leading cause of cancer-related deaths in worldwide, and the incidence of portal vein tumor thrombosis (PVTT), which is associated with dismal outcomes (median overall survival of 2.7–4 months), is 30–40% in patients with advanced HCC [1–3]. The development of radiotherapy technology facilitates the delivery of high doses to tumors with the minimal radiation doses to organs at risk (OARs), and radiotherapy has been increasingly applied to HCC with PVTT. Choi *et al.* demonstrated the feasibility outcome of the stereotactic body radiotherapy (30–39 Gy in 3 fractions) with the median survival period of eight months [4].

In radiotherapy, an application of four-dimensional computed tomography (4D-CT) is widely utilized to determine the position and respiratory motion of the target as well as OARs. However, the difference in CT numbers (Hounsfield unit, HU) between the liver and portal vein is small, and thus, it is difficult to visualize the HCC with PVTT clearly with the conventional 4D-CT. To overcome this problem, a methodology of contrast-enhanced 4D-CT (CE-4D-CT) was presented, and it could allow both enhancement of liver tumor contrast and coverage over the entire breathing cycle [5]. The CE-4D-CT has the potential for improving the accuracy of tumor contouring and localization.

Recently, dual-energy CT (DECT) has been increasingly introduced in clinical practice. By using high- and low-energy X rays, the DECT can reconstruct virtual monochromatic images (VMI) at different energy levels ranging 40–140 keV [6, 7]. Husarik *et al.* demonstrated a value of DECT (low image noise, high contrast-to-noise ratio (CNR), and high lesion conspicuity) for imaging of both hyperattenuating and hypoattenuating liver lesions [8]. Therefore, the CE-4D-CT using the DECT (CE-4D-DECT) has potential for improving the image quality of HCC with PVTT, and for measuring the respiration motion.

This study aimed to assess the objective and subjective image quality of the CE-4D-DECT for the HCC with PVTT for radiotherapy treatment planning. Further, the respiration motion of the PVTT was assessed and its motion was compared with that of the diaphragm.

## Materials and methods

### Acquisition of CE-4D-DECT

This study was approved by the Institutional Review Board of the Osaka International Cancer Institute (No. 18276), and written informed consent was waived because of the retrospective design. The CE-4D-DECT was performed for twelve patients (ten males and two females; age, 65 years (range, 48–78 years)) for radiotherapy treatment planning. In accordance with the Liver Cancer Study Group of Japan, each PVTT was classified into four categories [9]. Ten patients had PVTT with portal invasion at the first portal branch (Vp3), and two patients had PVTT with portal invasion at the main portal branch (Vp4).

Before simulation, patients' food or water intake was constrained to avoid unexpected displacement of PVTT due to a bulky stomach. The vacuum cushion was used for patient immobilization, and except for one case, a thermoplastic mold was mounted on the chest and abdomen to prevent a patient from breathing deeply. Thirty milliliters of water containing 5 ml of oral contrast medium was ingested, and a contrast agent (600 mgI/kg body weight) was injected in 30 s using a power injector.

The details of the acquisition technique of the CE-4D-DECT was presented in our previous study [10], and all CT images were acquired using a DECT scanner (Revolution HD, GE Medical Systems, Waukesha, WI) [11]. Briefly, the images of center of PVTT were acquired after 40–80 s from the injection, and the scan delay was changed depending on the location of the

tumor, patient's breathing cycle. All images were acquired in cine mode, and the cine duration time was set approximately 1 s longer than the patient's respiratory period. The Realtime Position Management system (Varian Medical Systems, Palo Alto, CA) was used to record the patient's respiratory waveform (free breathing) during the image acquisition. The tube voltage of 140/80 kVp, tube current of 360 mA, gantry rotation time of 0.5 s, and beam collimation of 40 mm was used for the image acquisition. The acquired data were reconstructed with a 2.5-mm slice thickness, a 512 × 512 matrix and a 500-mm field of view.

## Data analysis

The image data were transferred to a workstation (Advantage Sim, GE Medical Systems), and the end-exhale and end-inhale respiratory phase image sets were reconstructed using the patient's respiratory waveform. In the exhale phase image derived from the CE-4D-DECT in each patient, a circular or elliptic region of interest (ROI) was placed in the portal vein, tumor, aorta, liver, erector muscle of spine. For each ROI, the mean and standard deviation (SD) of the HU values were measured the VMIs at various energy levels in the range from 40keV to 140 keV, in 5 keV increments. The SD was determined as the image noise in the ROI. To evaluate the objective image quality of the VMI, the lesion contrast (LC) and the CNR between the portal vein and tumor were calculated using the following formulae:

$$LC[HU] = CT\ number\ in\ ROI_{portal\ vein} - CT\ number\ in\ ROI_{tumor}. \tag{1}$$

$$CNR = \frac{LC}{\sqrt{(Image\ noise\ in\ ROI_{portal\ vein})^2 + (Image\ noise\ in\ ROI_{tumor})^2}}. \tag{2}$$

The optimal VMI (O-VMI) for PVTT delineation for radiotherapy treatment planning was determined as the VMI, which has the highest CNR. The objective image characteristics of the O-VMI were compared with those of the standard VMI (S-VMI) at 77 keV, which shows equivalent HU values with the conventional single-energy CT images (120kVp) [12].

Subsequently, three radiation oncologists assessed the subjective image quality of the S-VMI and O-VMI with respect to the overall image quality and tumor conspicuity. Images were presented to the observers with the window width of 320 HU and window level of 30 HU to score the subjective image quality using the five-point scales (1, very poor; 2, Poor; 3, Satisfactory; 4, Good; 5, Very good). The observers determined the score from the viewpoint of contouring of the gross tumor volume (GTV) and/or OARs.

Using a treatment planning system (Eclipse, Varian Medical Systems), the respiratory motion of the tumor and diaphragm was assessed based on the O-VMIs at exhale and inhale phase (Fig 1). The tumor motion was determined as the difference in the position of the center of the GTV delineated by radiation oncologists between the two images in the left-right (L-R), anterior-posterior (A-P), and superior-inferior (S-I) directions, respectively. Radiation oncologists determined the GTV as the PVTT alone or PVTT combined with the primary tumor. The 3D motion was calculated as the root-mean-square of the tumor motion in the three directions. The diaphragm motion was determined as the difference in the position of the top of the liver dome delineated by a medical physicist between the two images in the three directions, and the 3D motion was also calculated.

The paired Wilcoxon signed-rank test (IBM SPSS Statistics version 24; IBM, Armonk, NY, USA) was used to measure the difference in objective and subjective analysis, and respiratory motion. A $p$ value of $<0.05$ was considered to indicate statistical significance.

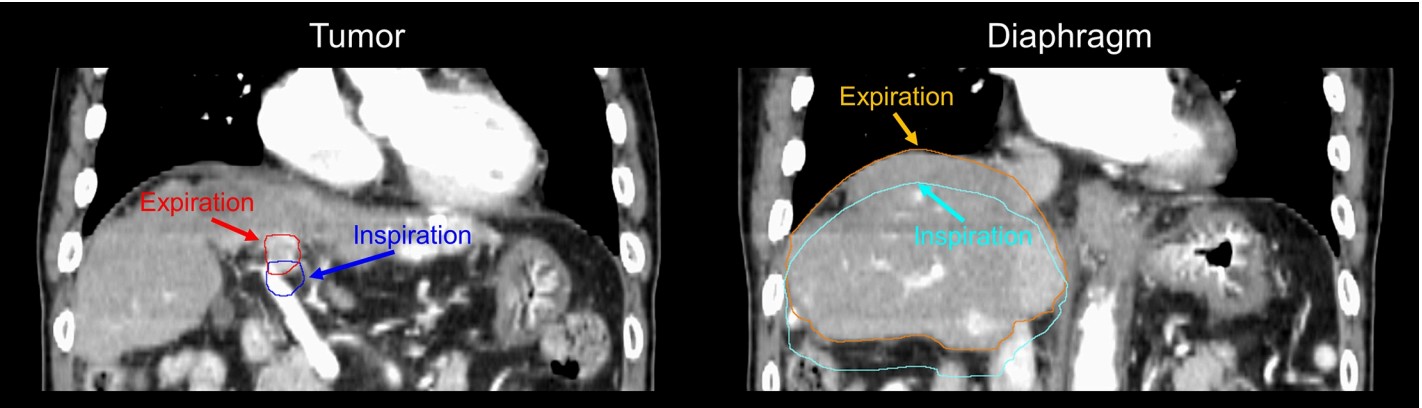

**Fig 1. Displacement in tumor and diaphragm positions between exhale and inhale phase images.**

## Results

Fig 2 shows the CT numbers and image noise in the portal vein and tumor, and the corresponding CNR at different energy levels. The higher HU values were observed in the low energy level of the VMI for both portal vein and tumor, and difference in the HU values between the VMI at low (40 keV) and high (140 keV) energy level was larger in the portal vein than that in the tumor (Fig 2A). The image noise was also larger at the low energy level on the VMI (Fig 2B). The resultant CNR was highest in the VMI at 60 keV, and thus, the image was determined as the O-VMI in this study (Fig 2C).

Fig 3 compares the objective quantitative values of the S-VMI with those of O-VMI. The HU values of the aorta (258.8 ± 113.8 HU), liver (113.8 ± 30.2 HU), muscle (59.8 ± 7.5 HU), portal vein (225.4 ± 49.8 HU), and tumor (79.8 ± 30.2 HU) in the O-VMI were significantly higher ($p = 0.002$) than the corresponding values in the S-VMI (158.7 ± 63.5, 86.5 ± 18.9, 53.1 ± 5.2, 141.1 ± 28.3, and 56.7 ± 18.8 HU for the aorta, liver, muscle, portal vein and tumor, respectively). In contrast, the HU value of the fat in the O-VMI (-97.7 ± 18.3 HU) was significantly lower ($p = 0.002$) than that in the S-VMI (-84.5 ± 13.4 HU). The O-VMI demonstrated the superiority of the objective image quality over the S-VMI, showing significantly higher LC (134.9 ± 43.6 HU vs 79.1 ± 26.2 HU, $p = 0.002$) and CNR (4.8 ± 1.4 vs 3.8 ± 1.2, $p = 0.002$). Fig 4 depicts the S-VMI and O-VMI for patient #3, 4, 6, and 7 in the axial and coronal view point. O-VMI provided better tumor conspicuity than the S-VMI, resulting in higher CRN for each patient (O-VMI vs S-VMI: 3.6 vs 5.0, 4.1 vs 4.8, 5.49 vs 7.4, and 3.8 vs 5.0 for patient #3, 4, 6, and 7, respectively.).

Table 1 summarizes the subjective image quality determined by three observers. Regarding the overall image quality, the O-VMI provided a significantly higher ($p = 0.003$) score (4.0 ± 0.6) than that in the S-VMI (3.1 ± 0.5). As in the tumor conspicuity, the mean score in the O-VMI was significantly higher ($p = 0.003$) than the corresponding value in the S-VMI (3.8 ± 0.8 and 2.8 ± 0.6 for O-VMI and S-VMI, respectively).

The individual respiratory motion between the tumor and diaphragm are directly compared in Fig 5. The diaphragm motion in the L-R (3.3 ± 2.5 mm, $p = 0.02$) and A-P (6.7 ± 4.0 mm, $p = 0.004$) directions were significantly larger compared to the tumor motion (1.2 ± 1.1 mm and 1.6 ± 1.3 mm in the L-R and A-P direction, respectively). The respiratory motion of the tumor (8.4 ± 3.6 mm) and diaphragm (10.1 ± 3.9 mm) were comparable in the S-I direction ($p = 0.3$). Consequently, the 3D motion of the diaphragm (13.1 ± 4.9 mm, $p = 0.03$) was significantly larger than the tumor motion (8.8 ± 3.5 mm).

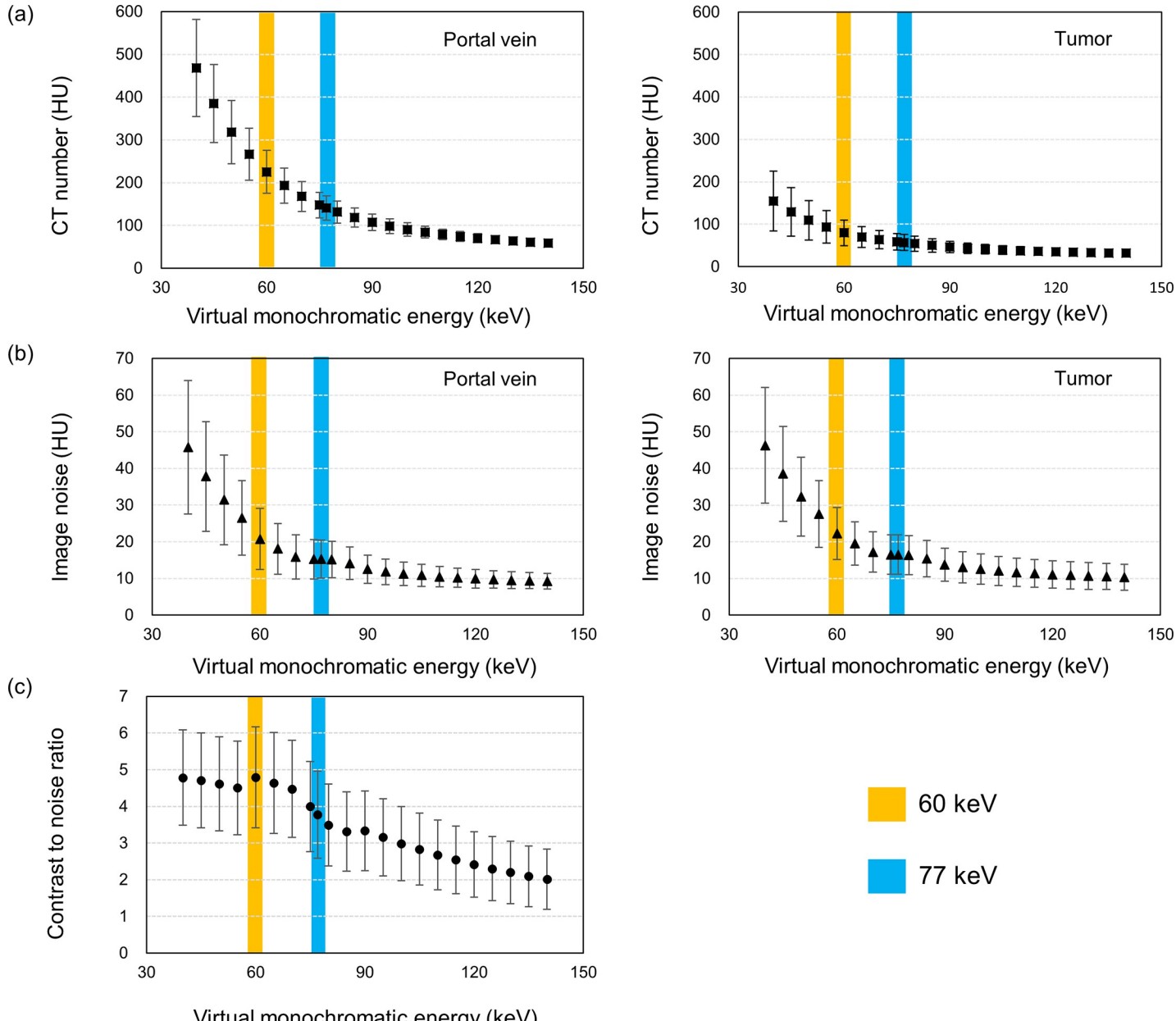

**Fig 2.** Quantitative spectral analysis of virtual monochromatic images at different energy levels; (a) the mean CT number and (b) image noise of the region of interest, and (c) contrast-to-noise ratio between portal vein and tumor. The error bar indicates the standard deviation.

## Discussion

In this study, we demonstrated the superiority of O-VMI (60 keV) over S-VMI (77 keV) in regard to the objective image quality, expressed as the CNR. Further, radiation oncologists judged that the O-VMI provided better overall image quality as well as tumor conspicuity.

Historically, the role of radiotherapy for the management of HCC has been limited due to the low radiation tolerance of the liver tissue. Kim *et al.* reported that the total liver volume receiving $\geq$ 30 Gy appears to be a useful dose–volumetric parameter for predicting the risk of Grade 2 (or worse) radiation-induced liver toxicity (RILT), and this volume should be limited

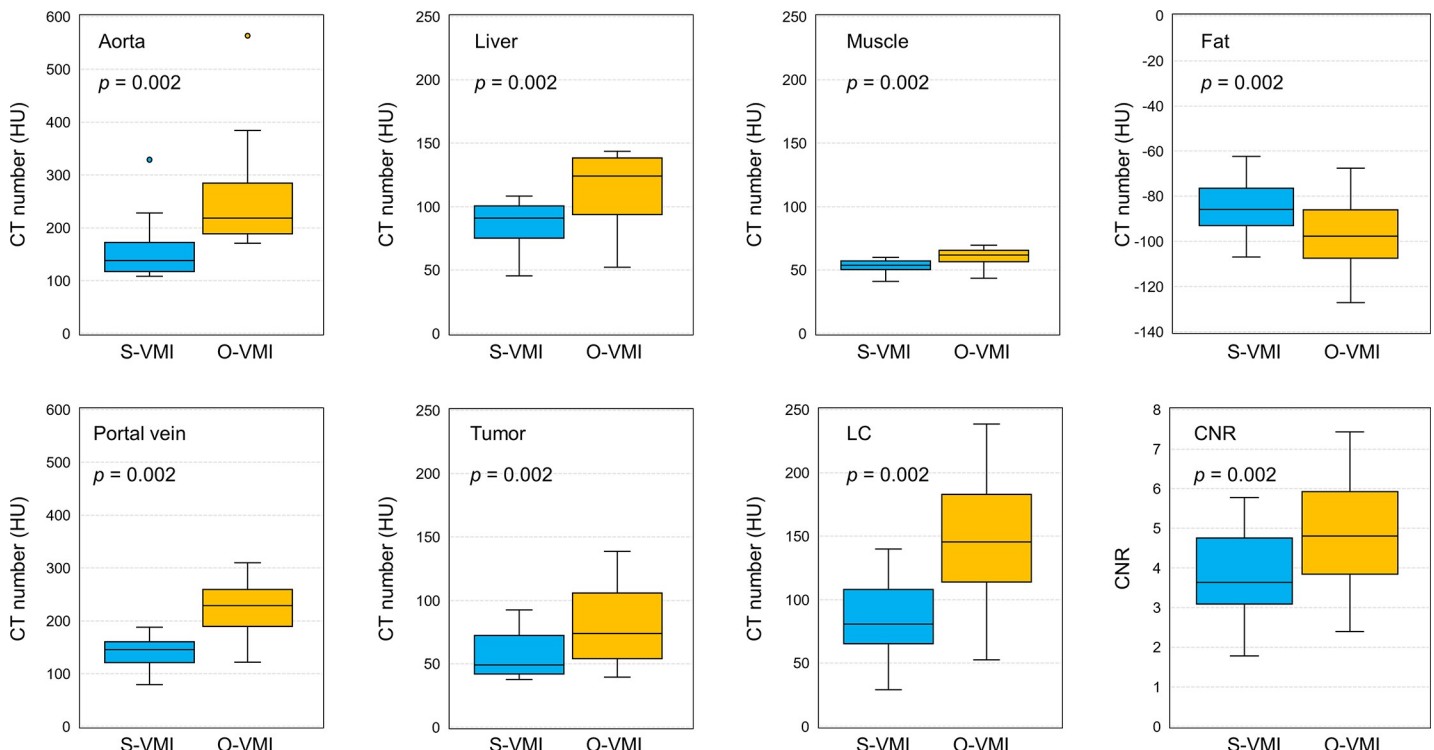

**Fig 3. Comparison of objective image quality between Standard Virtual Monochromatic Images (S-VMI, 77 keV) and optimal VMI (O-VMI, 60 keV).** Boxes, median value and upper and lower quartiles; Whiskers, maximum and minimum values within 1.5 × inter-quartile range; Dots, outliers.

to ≤ 60% whenever possible [13]. Dawson observed the 19 RILT among 203 patients who underwent liver irradiation, and no cases of RILT were observed when the mean liver dose was < 31 Gy [14]. Because of the better information regarding RILT, innovative RT technologies including intensity modulated radiotherapy, SBRT, particle therapy, and improvement in the precision of dose delivery with image-guided radiotherapy, radiotherapy has become an accepted tool in the management of HCC [1]. Shui *et al.* applied SBRT for 70 patients with PVTT, and the thrombus shrinkage and portal vein flow restoration could be achieved in the majority of cases [15]. Thus, they concluded that the SBRT can be used as the first-line therapy for HCC patients with extensive PVTT originally considered unsuitable for surgical resection or transarterial chemoembolization. For such treatments, precise and accurate determination of the moving target is required to minimize doses for normal liver tissues, because HCC with PVTT is associated with worse Child-Pugh score [16].

For target delineation in radiotherapy treatment planning, Beddar *et al.* demonstrated that the CE-4D-CT allowed both enhancement of liver tumor (colorectal liver metastases, cholangiocarcinoma, and hepatocellular carcinoma) contrast and coverage over the entire breathing cycle, and they concluded that the CE-4D-CT might improve the accuracy of tumor contouring and localization [5]. With regard to the imaging of PVTT, several applications were applied such as ultrasonography (US) and magnetic resonance imaging (MRI) [17, 18]. Rossi *et al.* demonstrated that CE-US detected 100% of PVTT while CE-CT detected 68% of PVTT [17]. Kim *et al.* demonstrated that the gadoxetic acid–enhanced MRI (GA-MRI) provided good diagnostic performance in the detection of PVTT with sensitivities of approximately 90% [18]. For the detection of PVTT using the GA-MRI, Bae *et al.* reported that the additional information of diffusion- and T2-weighted imaging, which are not available with CECT, could

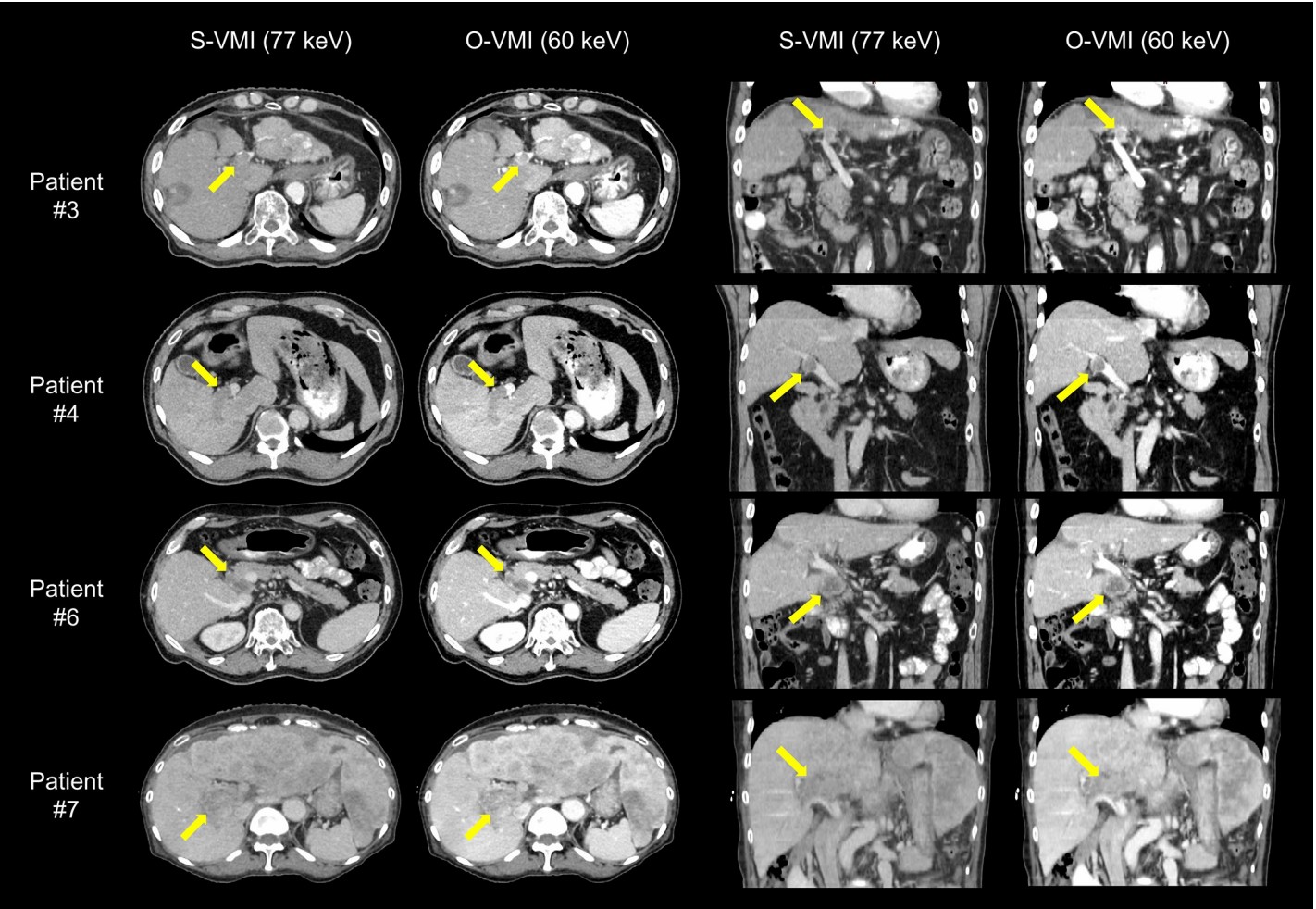

**Fig 4. Comparison of Standard Virtual Monochromatic Images (S-VMI) at 77 keV and optimal VMI (O-VMI) at 60 keV for patient #3, 4, 6, and 7.** Arrows indicate the portal vein tumor thrombosis.

contribute better depiction of common imaging findings, such as continuity with the tumor in the adjacent liver parenchyma than CE-CT [19], The study by Kim *et al.* demonstrated that the PVTT showed the increased T2 signal intensity and diffusion restriction, and the characteristic imaging features improved diagnostic capability [18]. Moreover, the diffusion-weighted imaging has advantage for distinguishing bland thrombus from neoplastic thrombus in the portal vein in patients with HCC [20]. Although the conventional CE-CT has been widely used for first-line diagnostic tests in patients suspected of having HCC, there is an apparent necessity for the improvement of image quality in CE-CT images. This study first implemented the CE-4D-DECT for PVTT and the imaging technique could provide VMIs at various energy level for the moving target. In this study, the VMI at 60 keV provided the highest CNR, resulting in higher subjective quantitative values than the S-VMI. Similar results were reported by Shuman *et al.* that the VMI at 50 keV showed greater CNR and higher subjective quantitative values for patients with HCC than the corresponding values of the VMI at 77 keV [21]. On the basis of conventional CE-4D-CT scans, Jensen *et al.* reported that there is non-negligible interobserver variability in HCC delineation in radiotherapy treatment planning, and they suggested the need for improving the image quality the CE-4D-CT [22]. The improvement of the image

**Table 1. Subjective image quality expressed as five-point sores.**

| Patient # | Overall image quality | | | | | | Tumor conspicuity | | | | | |
|---|---|---|---|---|---|---|---|---|---|---|---|---|
| | Observer 1 | | Observer 2 | | Observer 3 | | Observer 1 | | Observer 2 | | Observer 3 | |
| | S-VMI | O-VMI | S-VMI | O-VMI | S-VMI | O-VMI | S-VMI | O-VMI | S-VMI | O-VMI | S-VMI | O-VMI |
| 1 | 3 | 4 | 3 | 4 | 2 | 3 | 3 | 4 | 2 | 3 | 2 | 3 |
| 2 | 3 | 3 | 3 | 3 | 2 | 3 | 3 | 3 | 1 | 3 | 2 | 2 |
| 3 | 3 | 4 | 4 | 4 | 3 | 4 | 3 | 4 | 3 | 4 | 3 | 4 |
| 4 | 3 | 4 | 3 | 4 | 3 | 4 | 3 | 4 | 3 | 5 | 3 | 4 |
| 5 | 3 | 4 | 3 | 3 | 3 | 4 | 3 | 4 | 2 | 4 | 3 | 4 |
| 6 | 3 | 4 | 4 | 5 | 4 | 4 | 3 | 4 | 3 | 4 | 4 | 4 |
| 7 | 3 | 4 | 3 | 5 | 3 | 5 | 3 | 4 | 3 | 4 | 3 | 4 |
| 8 | 3 | 4 | 3 | 5 | 2 | 4 | 3 | 4 | 3 | 5 | 3 | 4 |
| 9 | 3 | 4 | 4 | 5 | 4 | 4 | 3 | 4 | 4 | 5 | 3 | 4 |
| 10 | 4 | 4 | 4 | 5 | 3 | 4 | 3 | 4 | 3 | 5 | 3 | 4 |
| 11 | 3 | 4 | 3 | 4 | 3 | 4 | 2 | 3 | 3 | 3 | 2 | 3 |
| 12 | 3 | 3 | 3 | 4 | 4 | 3 | 3 | 4 | 2 | 2 | 3 | 2 |
| Mean | 3.1 | 3.8 | 3.3 | 4.3 | 3.0 | 3.8 | 2.9 | 3.8 | 2.7 | 3.9 | 2.8 | 3.5 |
| SD | 0.3 | 0.4 | 0.5 | 0.8 | 0.7 | 0.6 | 0.3 | 0.4 | 0.8 | 1.0 | 0.6 | 0.8 |
| Overall mean (S-VMI vs O-VMI) | 3.1 vs 4.0 | | | | | | 2.8 vs 3.8 | | | | | |
| $p$-value | 0.003 | | | | | | 0.003 | | | | | |

quality using the CE-4D-DECT may reduce the interobserver variability in PTVV delineation over the respiratory cycle in radiotherapy treatment planning.

Fernandes *et al*. assessed the liver tumor motion (hepatocellular carcinoma ($n = 11$), cholangiocarcinoma ($n = 3$), and liver metastasis ($n = 2$)) using the 4D-CT, and they observed the mean respiratory motion of 2.1, 4.8 8.0 mm in the L-R, A-P and S-I directions [23]. Our assessment of the respiratory motion of the PVTT yielded the similar results by Fernandes *et al*. that the respiratory motion was largest in the S-I direction. The motion of the PVTT was significantly smaller than the liver motion, which implied the motion of diaphragm could not well represent that of PVTT, especially in the L-R and A-P directions (Fig 5). Yang *et al*. reported that the tumor and diaphragm motions had high concordance when the distance between the tumor and tracked diaphragm area was small [24]. Because the distance between the diaphragm and PVTT is relatively large, the fiducial marker might be required to surrogate the PVTT when high radiation dose is delivered with a tight margin.

Several limitations of this study warrant mention. First, our data could not support all categories of the classification of PVTT (Vp1-Vp4) and various tumor sizes. A small PVTT can be easily missed by CT, which is unable to distinguish between tumor and thrombus tissues [17]. Second, CE-4D-DECT can be acquired at only one respiratory phase (40–80 s after the injection in this study) while diagnostic CT images are commonly acquired in four phases (pre-contrast, arterial, portal, and late). Finally, the objective and subjective image qualities between the O-VMI and the S-VMI were compared in this study, and the corresponding values of the conventional breath-hold polychromatic image (120 kVp) could not be assessed.

In conclusion, the VMI at 60 keV can be considered as the optimal image for radiotherapy treatment planning because that provided the highest CNR between the tumor and portal vein. Further, the optimal VMI significantly improved subjective image quality assessed by radiation oncologists. Because the respiratory motion of the diaphragm was significantly larger than that of the PVTT in the L-R and A-P directions, we need to be pay attention for localizing target in radiotherapy.

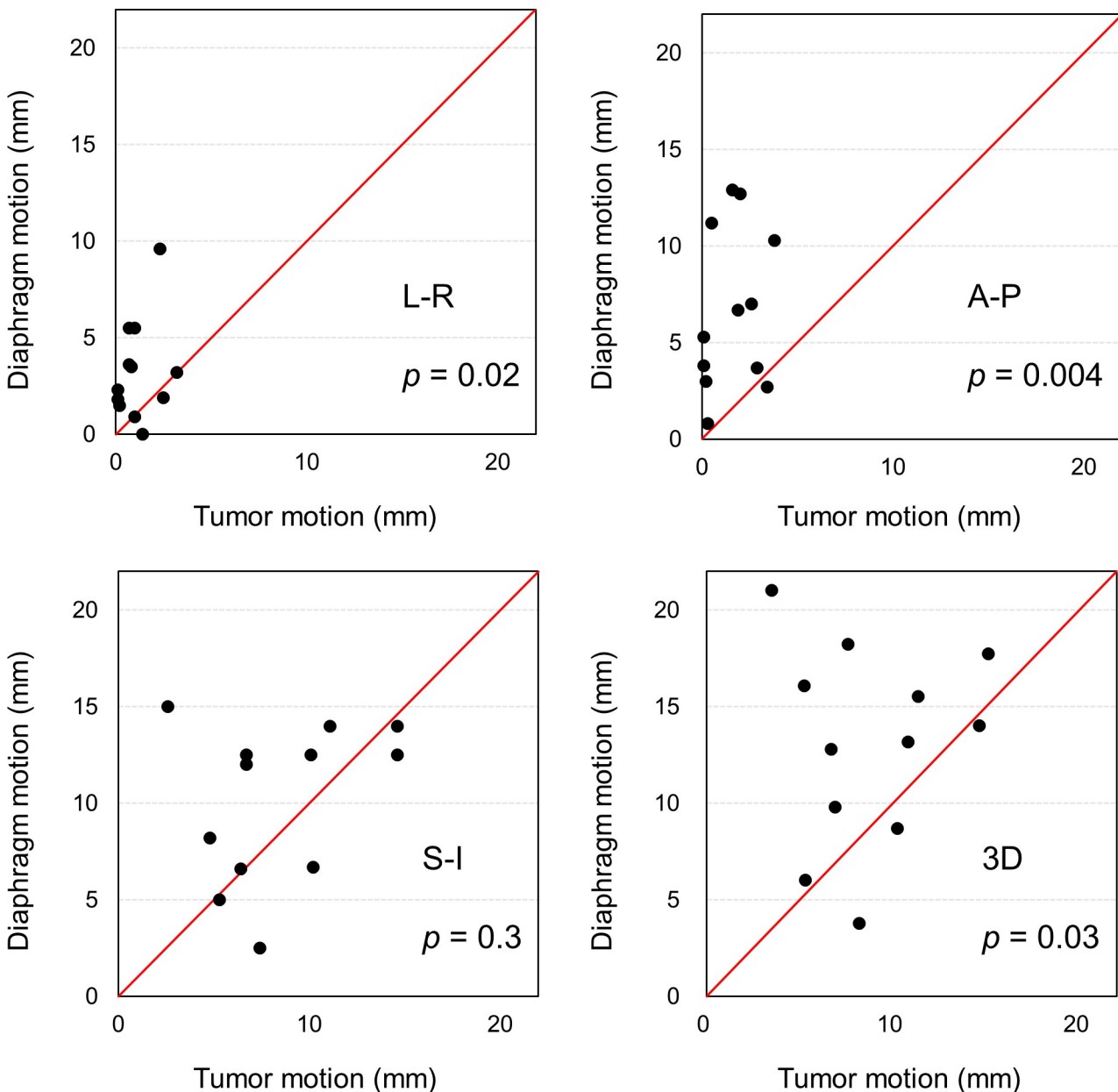

**Fig 5. Distribution of respiratory motion of tumor (horizontal axis) and diaphragm (vertical axis) for individual patients.**

## Author Contributions

**Conceptualization:** Shingo Ohira.

**Data curation:** Shingo Ohira, Kentaro Wada, Toshiki Ikawa, Takero Hirata.

**Formal analysis:** Shingo Ohira, Kentaro Wada, Takero Hirata.

**Funding acquisition:** Shingo Ohira.

**Investigation:** Shingo Ohira, Naoyuki Kanayama, Kentaro Wada, Toshiki Ikawa, Takero Hirata, Tsukasa Karino, Hayate Washio, Yoshihiro Ueda, Masayoshi Miyazaki.

**Methodology:** Shingo Ohira, Naoyuki Kanayama, Kentaro Wada, Toshiki Ikawa, Takero Hirata, Noriko Kishi, Tsukasa Karino, Hayate Washio, Yoshihiro Ueda, Masayoshi Miyazaki.

**Project administration:** Shingo Ohira, Naoyuki Kanayama, Kentaro Wada, Toshiki Ikawa.

**Resources:** Shingo Ohira.

**Software:** Shingo Ohira.

**Supervision:** Shingo Ohira, Masayoshi Miyazaki, Masahiko Koizumi, Teruki Teshima.

**Validation:** Shingo Ohira, Yoshihiro Ueda.

**Visualization:** Shingo Ohira, Yoshihiro Ueda.

**Writing – original draft:** Shingo Ohira, Yoshihiro Ueda.

**Writing – review & editing:** Shingo Ohira, Tsukasa Karino, Hayate Washio, Yoshihiro Ueda, Teruki Teshima.

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
