## [Decision Letter · Decision Letter 0]

1 Sep 2020

PONE-D-19-33675

Improvement of image quality and assessment of respiratory motion for hepatocellular carcinoma with portal vein tumor thrombosis using contrast-enhanced four-dimensional dual-energy computed tomography

PLOS ONE

Dear Dr. Schingo Ohira,

Thank you for submitting your manuscript to PLOS ONE. After careful consideration, we feel that it has merit but does not fully meet PLOS ONE’s publication criteria as it currently stands. Therefore, we invite you to submit a revised version of the manuscript that addresses the points raised during the review process.

We look forward to receiving your revised manuscript.

Kind regards,

Agostino Chiaravalloti, MD, PhD

Academic Editor

PLOS ONE

Journal Requirements:

2.Thank you for including your ethics statement:  "This study was approved by our ethics committee, and written informed consent was obtained from each patient."

3.Thank you for stating the following in the Acknowledgments Section of your manuscript:

[This study was supported by JSPS KAKENHI Grant (Grant-in-Aid for Young

Scientists 19K17285).]

 [The funders had no role in study design, data collection and analysis, decision to publish, or preparation of the manuscript.]

Additional Editor Comments (if provided):

Reviewers have now commented on your paper. You will see that they are advising that you revise in a minor way your manuscript.

If you are prepared to undertake the work required, I could reconsider my decision.

Reviewers' comments:

Reviewer's Responses to Questions

**Comments to the Author**

1. Is the manuscript technically sound, and do the data support the conclusions?

Reviewer #1: Yes

2. Has the statistical analysis been performed appropriately and rigorously? 

Reviewer #1: Yes

3. Have the authors made all data underlying the findings in their manuscript fully available?

Reviewer #1: Yes

4. Is the manuscript presented in an intelligible fashion and written in standard English?

Reviewer #1: Yes

5. Review Comments to the Author

Reviewer #1: Overall, this research was well-designed. It suggested feasibility to used the CE-4D-DECT for better delineation of hepatocellular carcinoma with portal vein thrombosis. Authors have mentioned the limitation in phase of enhancement, which I agreed. Visibility of primary hepatocellular carcinoma and portal vein thrombosis is better in combined phases and the patterns of enhancement are various.

Movement of tumor correlated with diaphragm is also point of concerned. Authors point out the discordance between diaphragm and tumor movement.

I agreed but it would be better understand if authors would described more about how to measure tumor movement, eg.

The definition of GTV; Primary alone or combined with PVTT?

How to find the center of GTV? Was it contoured by radiation oncologist?

It would be appreciate if author have described it with figure, might be in supplement data.

In discussion, it would be outright if authors would add a paragraph mention about limitation of previously used or conventional technique in target delineation of hepatocellular carcinoma with some evidences.

6. PLOS authors have the option to publish the peer review history of their article (what does this mean?). If published, this will include your full peer review and any attached files.

Reviewer #1: No

---

## [Author Response · Author response to Decision Letter 0]

10 Sep 2020

Dear Editor and Reviewers,

We would like to thank you for the insightful comments on our paper. We have revised the manuscript in the light of the comments of the editor and reviewers. The revisions have been made using the 'Edit' function of Word. We hope that the revised version of our manuscript is now suitable for publication in the PLOS ONE.

Point-by-point responses according to the reviewers’ comments are as follows:

Editor’s Comments

Reviewers have now commented on your paper. You will see that they are advising that you revise in a minor way your manuscript.

If you are prepared to undertake the work required, I could reconsider my decision.

We have revised the manuscript in accordance with the reviewer’s comments. We hope that the revised version of our manuscript is now suitable for publication in the PLOS ONE.

Reviewer’s Comments

Reviewer #1: Overall, this research was well-designed. It suggested feasibility to used the CE-4D-DECT for better delineation of hepatocellular carcinoma with portal vein thrombosis. Authors have mentioned the limitation in phase of enhancement, which I agreed. Visibility of primary hepatocellular carcinoma and portal vein thrombosis is better in combined phases and the patterns of enhancement are various. Movement of tumor correlated with diaphragm is also point of concerned. Authors point out the discordance between diaphragm and tumor movement.

Thank you so much for your favorable comments. We have revised the manuscript in accordance with the reviewer’s comments. We hope that the revised version of our manuscript is now suitable for publication in the PLOS ONE.

I agreed but it would be better understand if authors would described more about how to measure tumor movement, eg.

The definition of GTV; Primary alone or combined with PVTT?

How to find the center of GTV? Was it contoured by radiation oncologist?

It would be appreciate if author have described it with figure, might be in supplement data.

We agree the reviewer’s comments, and the additional Figure was described to explain the tumor/diaphragm motion measurement. The tumor and diaphragm were delineated by radiation oncologists and a medical physicist, respectively. Radiation oncologists determined the GTV as the PVTT alone or PVTT combined with the primary tumor. The center of the tumor was calculated by using a treatment planning system (Eclipse, Varian Medical Systems). We believe that the readers can understand the methodology of the tumor/diaphragm motion in the revised manuscript. Thank you for your insightful comments.

In discussion, it would be outright if authors would add a paragraph mention about limitation of previously used or conventional technique in target delineation of hepatocellular carcinoma with some evidences.

We agree the reviewer’s comment. Although the CE-CT has been widely used for first-line diagnostic tests in patients suspected of having HCC, many major guidelines (such as European Association for the Study of the Liver and American Association for the Study of Liver Diseases) now include gadoxetic acid-enhanced MRI (GA-MRI) as a first-line diagnostic test. The limitation of conventional CE-CT is that conventional CT can only depict PTVV in the HU values. For the detection of PVTT using the GA-MRI, Bae et al. reported that the additional information of diffusion- and T2-weighted imaging could contribute better depiction of common imaging findings, such as continuity with the tumor in the adjacent liver parenchyma than CE-CT [Bae et al., Liver Cancer, 2020]. Previous study Kim et al demonstrated that the PVTT showed the increased T2 signal intensity and diffusion restriction, and the characteristic imaging features improved diagnostic capability [Kim et al., Radiology, 2016]. Moreover, the diffusion-weighted imaging has advantage for distinguishing bland thrombus from neoplastic thrombus in the portal vein in patients with HCC [Catalano et al., Radiology, 2010]. The evidences in detecting PVTT using MR imaging are added in the paragraph mentioned about limitation of previously used or conventional technique.

---

## [Editor Report · Decision Letter 1]

3 Dec 2020

Improvement of image quality and assessment of respiratory motion for hepatocellular carcinoma with portal vein tumor thrombosis using contrast-enhanced four-dimensional dual-energy computed tomography

PONE-D-19-33675R1

Dear Dr. Ohira,

We’re pleased to inform you that your manuscript has been judged scientifically suitable for publication and will be formally accepted for publication once it meets all outstanding technical requirements.

Kind regards,

Dong-Hoon Lee, Ph.D.

Academic Editor

PLOS ONE

Additional Editor Comments (optional):

Reviewers have now commented on your paper. You will see that they are advising that you revise in a minor way your manuscript.

If you are prepared to undertake the work required, I could reconsider my decision.
---

## [Editor Report · Acceptance letter]

22 Dec 2020

PONE-D-19-33675R1 

Improvement of image quality and assessment of respiratory motion for hepatocellular carcinoma with portal vein tumor thrombosis using contrast-enhanced four-dimensional dual-energy computed tomography 

Dear Dr. Ohira:

I'm pleased to inform you that your manuscript has been deemed suitable for publication in PLOS ONE. Congratulations! Your manuscript is now with our production department. 

Kind regards, 

on behalf of

Prof. Dong-Hoon Lee 

Academic Editor

PLOS ONE